# PRIVATE MULTI-TASK LEARNING: FORMULATION AND APPLICATIONS TO FEDERATED LEARNING

## ABSTRACT

Many problems in machine learning rely on *multi-task learning (MTL)*, in which the goal is to solve multiple related machine learning tasks simultaneously. MTL is particularly relevant for privacy-sensitive applications in areas such as healthcare, finance, and IoT computing, where sensitive data from multiple, varied sources are shared for the purpose of learning. In this work, we formalize notions of task-level privacy for MTL via *joint differential privacy* (JDP), a relaxation of differential privacy for mechanism design and distributed optimization. We then propose an algorithm for mean-regularized MTL, an objective commonly used for applications in personalized federated learning, subject to JDP. We analyze our objective and solver, providing certifiable guarantees on both privacy and utility. Empirically, we find that our method allows for improved privacy/utility trade-offs relative to global baselines across common federated learning benchmarks.

## 1 INTRODUCTION

Multi-task learning (MTL) aims to solve multiple learning tasks simultaneously while exploiting similarities/differences across tasks (Caruana, 1997). Multi-task learning is commonly used in applications that warrant strong privacy guarantees. For example, MTL has been used in healthcare, as a way to learn over diverse populations or between multiple institutions (Baytas et al., 2016; Suresh et al., 2018; Harutyunyan et al., 2019); in financial forecasting, to combine knowledge from multiple indicators or across organizations (Ghosn & Bengio, 1997; Cheng et al., 2020); and in IoT computing, as an approach for learning in federated networks of heterogeneous devices (Smith et al., 2017; Hanzely & Richtárik, 2020; Hanzely et al., 2020; Ghosh et al., 2020; Sattler et al., 2020; Deng et al., 2020; Mansour et al., 2020). While MTL can significantly improve accuracy when learning in these applications, there is a dearth of work studying the privacy implications of multi-task learning.

In this work, we develop and theoretically analyze methods for MTL with formal privacy guarantees. Motivated by applications in federated learning, we aim to provide *task-level privacy*[1], where each task corresponds to a client/device/data silo, and the goal is to protect the sensitive information in each task's data (McMahan et al., 2018). We focus on incorporating *differential privacy* (DP) (Dwork et al., 2006), which (informally) requires an algorithm's output to be insensitive to the change of any single entity's data. For MTL, using task-level DP directly would require the entire set of predictive models across all tasks to be insensitive to changes in the private data of any single task. This requirement is too stringent for most applications, as it implies that the predictive model for task $k$ must have little dependence on the training data for task $k$, thus preventing the usefulness of the model (see Figure 1).

To circumvent this limitation, we leverage a meaningful relaxation of DP known as *joint differential privacy* (JDP) (Kearns et al., 2014), which requires that for each task $k$, the set of output predictive models for all other tasks *except* $k$ is insensitive to $k$'s private data. As a consequence, the client's private data in task $k$ is protected even if all other clients/tasks collude and share their private data and output models (as long as client $k$ keeps their data private). In contrast to standard DP, JDP allows the predictive model for task $k$ to depend on $k$'s private data, helping to preserve the task's utility.

Using JDP, we then develop new learning algorithms for MTL with rigorous privacy and utility guarantees. Specifically, we propose Private Mean-Regularized MTL, a simple framework for learning

---

[1]In federated learning applications, where each MTL task typically corresponds to a client's local training task, this can be equivalently viewed as 'client-level' or 'user-level' privacy (McMahan et al., 2018).

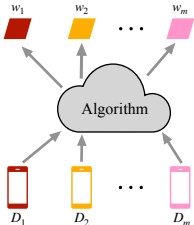 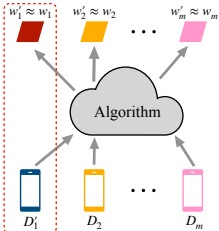 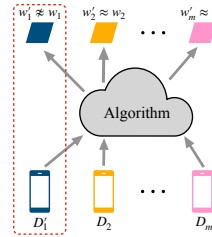

Figure 1: An MTL problem consists of $m$ different tasks and a learning algorithm that jointly produces one model for each task (Left). For example, in cross-device federated learning, each 'task' may represent data from a mobile phone client (as depicted), and MTL can be used to learn shared, yet personalized models for each client (Smith et al., 2017). In traditional differential privacy, if the private data of task $k$ (e.g., Task 1) changes, the models produced by the MTL algorithm should be indistinguishable from the models derived without changing data from task $k$ (Middle). In contrast, JDP allows the model of task $k$ to be dependent on task $k$'s data while still protecting other tasks from leaking information about their private data (Right).

multiple tasks while ensuring task-level privacy. We show that our method achieves $(\epsilon, \delta)$-JDP. Our scalable solver builds on FedAvg (McMahan et al., 2017), a common method for communication-efficient federated optimization. We analyze the convergence of our solver on both nonconvex and convex objectives, demonstrating a tradeoff between privacy and utility, and evaluate this trade-off empirically on multiple federated learning benchmarks. We summarize our contributions below:

- Our work is the first we are aware of to provide **formal definitions of task-level differential privacy for multi-task learning objectives** (Section 3). Our definitions rely on joint differential privacy and are applicable to commonly-used multi-task relationship learning objectives.

- Using our privacy definitions, we propose Private Mean-Regularized MTL, a simple MTL framework that provides task-level privacy (Section 4). We prove that our method achieves $(\epsilon, \delta)$-JDP, and we analyze the convergence of our communication-efficient solver on convex and nonconvex objectives. Our convergence analysis extends to non-private settings with partial participation, which may be of independent interest for problems in cross-device federated learning.

- Finally, we explore the performance of our approach on common federated learning benchmarks (Section 5). Our results show that it is possible to retain the accuracy benefits of MTL in these settings relative to global baselines while still providing meaningful privacy guarantees. Further, even in cases where the MTL objective achieves similar accuracy to the global objective, we find that privacy/utility benefits exist when employing the private MTL formulation.

## 2 BACKGROUND AND RELATED WORK

**Multi-task learning.** Multi-task learning considers jointly solving multiple related ML tasks. Our work focuses on the general and widely-used formulation of multi-task relationship learning (Zhang & Yeung, 2010), as discussed in Section 3. This form of MTL is particularly useful in privacy-sensitive applications where datasets are shared among multiple heterogeneous entities (Baytas et al., 2016; Smith et al., 2017; Ghosn & Bengio, 1997). In these cases, it is natural to view each data source (e.g., financial institution, hospital, mobile phone) as a separate 'task' that is learned in unison with the other tasks. This allows data to be shared, but the models to be personalized to each data silo. For example, in the setting of cross-device federated learning, MTL is commonly used to train a personalized model for each device in a distributed network (Smith et al., 2017; Liu et al., 2017).

**Federated learning.** A motivation for our work is the application of federated learning (FL), in which the goal is to collaboratively learn from data that has been generated by, and resides on, a number of private data silos, such as remote devices or servers (McMahan et al., 2017; Kairouz et al., 2019; Li et al., 2020a). To ensure client-level differential privacy in FL, a common technique is to learn one *global model* across the distributed data and then add noise to the aggregated model to sufficiently mask any specific client's update (Kairouz et al., 2019; McMahan et al., 2018; Geyer et al., 2017). However, a defining characteristic of federated learning is that the distributed data are likely to be heterogeneous, i.e., each client may generate data via a distinct data distribution (Kairouz et al., 2019; Li et al., 2020a). To model the (possibly) varying data distributions on each client, it is natural to instead consider learning a separate model for each client's local dataset.

To this end, a number of recent works have explored multi-task learning as a way to improve the accuracy of learning in federated networks (Smith et al., 2017; Hanzely & Richtárik, 2020; Hanzely

et al., 2020; Ghosh et al., 2020; Sattler et al., 2020; Deng et al., 2020; Mansour et al., 2020). *Despite the prevalence of multi-task federated learning, we are unaware of any work that has explored task-level privacy for commonly-used multi-task relationship models (Section 3) in federated settings.*

**Differentially private MTL.** Prior work in private MTL differs from our own either in terms of the privacy formulation or MTL objective. For example, Wu et al. (2020) explore a specific MTL setting where a feature representation shared by all tasks is first learned, followed by task-specific models on top of this private representation. We instead study multi-task relationship learning (Section 3), which is a general and widely-used MTL framework, particularly in federated learning (Smith et al., 2017). While our work focuses on task-level privacy, there has been work on data-level privacy for MTL, which aims to protect any single piece of local data rather than protecting the entire local dataset. For example, Xie et al. (2017) propose a method for data-level privacy by representing the model for each task as a sum of a public, shared weight and a task-specific weight that is only updated locally, and Gupta et al. (2016) study data-level privacy for a mean estimation MTL problem. Finally, Li et al. (2019) studies multiple notions of differential privacy for meta-learning. Although similarly motivated by personalization, their framework does not cover the multi-task setting, where there exists a separate model for each task.

## 3 MULTI-TASK LEARNING AND PRIVACY FORMULATION

In this section, we first formalize our multi-task learning objective, which is a form of mean-regularized multi-task learning (Section 3.1), and then provide our privacy formulation (Section 3.2).

### 3.1 PROBLEM SETUP

In the classical setting of multi-task relationship learning (Zhang & Yang, 2017; Zhang & Yeung, 2010), there are $m$ different task learners with their own task-specific data. The aim is to solve:

$$\min_{W,\Omega} \left\{ F(W,\Omega) = \left\{ \frac{1}{m} \sum_{k=1}^{m} \sum_{i=1}^{n_k} l_k(x_i, w_k) + \mathcal{R}(W,\Omega) \right\} \right\}, \tag{1}$$

where $w_k$ is model for task learner $k$; $\{x_1, \ldots, x_{n_k}\}$ is the local data for the $k^{th}$ task; $l_k(\cdot)$ is the empirical loss for task $k$; $W = [w_1; \cdots; w_m]$; and $\Omega \in \mathbb{R}^{m \times m}$ characterizes the relationship between every pair of task learners. A common choice for setting the regularization term $\mathcal{R}(W,\Omega)$ in previous works (Zhang & Yeung, 2010; Smith et al., 2017) is:

$$\mathcal{R}(W,\Omega) = \lambda_1 \mathrm{tr}(W\Omega W^T),$$

where $\Omega$ can be viewed as a covariance matrix, used to learn/encode positive, negative, or unrelated task relationships (Zhang & Yeung, 2010). In this paper, we focus on studying the mean-regularized multi-task learning objective (Evgeniou & Pontil, 2004): a special case of (1) where $\Omega = (\mathbf{I_{m \times m}} - \frac{1}{\mathbf{m}} \mathbf{1_m} \mathbf{1_m^T})^2$ is fixed. Here $\mathbf{I_{m \times m}}$ is the identity matrix of size $m \times m$ and $\mathbf{1_m} \in \mathbb{R}^{\mathbf{m}}$ is the vector with all entries equal to 1. By picking $\lambda_1 = \frac{\lambda}{2}$, we can rewrite the objective as:

$$\min_{W} \left\{ F(W) = \left\{ \frac{1}{m} \sum_{k=1}^{m} \frac{\lambda}{2} \|w_k - \bar{w}\|^2 + \sum_{i=1}^{n_k} l_k(x_i, w_k) \right\} \right\}, \tag{2}$$

where $\bar{w}$ is the average of task-specific models: $\bar{w} = \frac{1}{m} \sum_{i=1}^{m} w_k$. Note that $\bar{w}$ is shared across all tasks, and each $w_k$ is kept locally for task learner $k$. During optimization, each task learner $k$ solves:

$$\min_{w_k} \left\{ f_k(w_k; \bar{w}) = \frac{\lambda}{2} \|w_k - \bar{w}\|^2 + \sum_{i=1}^{n_k} l_k(x_i, w_k) \right\}. \tag{3}$$

Despite the prevalence of this simple form of multi-task learning and its recent use in applications such as federated learning with strong privacy motivations (e.g., Hanzely & Richtárik, 2020; Hanzely et al., 2020; Dinh et al., 2020), we are unaware of prior work that has formalized task-level differential privacy in the context of solving Objective (2).

## 3.2 Privacy Formulation

We start by introducing the definition of *differential privacy (DP)* before discussing its generalization to *joint differential privacy (JDP)*. In the context of multi-task learning, each of the $m$ task learners owns a private dataset $D_i \in \mathcal{U}_i \subset \mathcal{U}$. We define $D = \{D_1, \cdots, D_m\}$ and $D' = \{D'_1, \cdots, D'_m\}$. We call two sets $D, D'$ *neighboring sets* if they only differ on the index $i$, i.e., $D_j = D'_j$ for all $j$ except $i$. With this setup in mind, we define differential privacy more formally below.

**Definition 1** (Differential Privacy (DP) for MTL (Dwork et al., 2006)). *A randomized algorithm* $\mathcal{M} : \mathcal{U}^m \rightarrow \mathcal{R}^m$ *is* $(\epsilon, \delta)$*-differentially private if for every pair of neighboring sets that only differ in arbitrary index* $i$: $D, D' \in \mathcal{U}$ *and for every set of subsets of outputs* $S \subset \mathcal{R}$,

$$Pr(\mathcal{M}(D) \in S) \leq e^\epsilon Pr(\mathcal{M}(D') \in S) + \delta. \tag{4}$$

In the context of MTL, a learning algorithm outputs one model for every task learner. As mentioned previously, since the output of MTL is a *collection of models*, traditional DP would require that all the models produced by an MTL learning algorithm are insensitive to changes that happen to the private dataset of *any* single task.

In this work we are interested in studying task-level privacy, where the purpose is to protect one task learner's data from leakage to any other task learners. In this setting, DP incurs an additional restriction that the model of any task learner should also be insensitive to changes in *its own data*, which would render each of the models useless. To overcome this limitation of DP, we suggest employing *joint differential privacy (JDP)* (Kearns et al., 2014), a relaxed notion of DP, to formalize the guarantee that an MTL algorithm should provide in order to protect task-level privacy. Intuitively, JDP requires that for each task $k$, the set of output predictive models for all other tasks **except** $k$ is insensitive to $k$'s private data. We provide a formal definition below.

**Definition 2** (Joint Differential Privacy (JDP) (Kearns et al., 2014)). *A randomized algorithm* $\mathcal{M} : \mathcal{U}^m \rightarrow \mathcal{R}^m$ *is* $(\epsilon, \delta)$*-joint differentially private if for every* $i$, *for every pair of neighboring datasets that only differ in index* $i$: $D, D' \in \mathcal{U}^m$ *and for every set of subsets of outputs* $S \subset \mathcal{R}^m$,

$$Pr(\mathcal{M}(D)_{-i} \in S) \leq e^\epsilon Pr(\mathcal{M}(D')_{-i} \in S) + \delta, \tag{5}$$

*where* $\mathcal{M}(D)_{-i}$ *represents the vector* $\mathcal{M}(D)$ *with the* $i$-*th entry removed.*

JDP allows the predictive model for task $k$ to depend on the private data of $k$, while still providing a strong guarantee: even if all the clients from all the other tasks collude and share their information, they still will not be able to learn much about the private data in the task $k$. JDP has mostly been used in applications related to mechanism design (Hsu et al., 2016a; Kannan et al., 2015; Hsu et al., 2016b; Cummings et al., 2015; Rogers & Roth, 2014; Kearns et al., 2014). Although it is a natural choice for achieving task-level privacy in MTL, we are unaware of any work that studies MTL subject to JDP.

We also note that we can naturally connect joint differential privacy to standard differential privacy. Informally, if we take the output of a differentially private process and run some algorithm on top of that locally for each task learner *without* communicating to the global learner or other task learners, this whole process can be shown to be joint differentially private. This is formalized as the *Billboard Lemma* (Kearns et al., 2014), presented in Lemma 1 below.

**Lemma 1** (Billboard Lemma). *Suppose* $\mathcal{M} : \mathcal{D} \rightarrow \mathcal{W}$ *is* $(\epsilon, \delta)$*-differentially private. Consider any set of functions:* $f_i : \mathcal{D}_i \times \mathcal{W} \rightarrow \mathcal{W}'$. *The composition* $\{f_i(\Pi_i \mathcal{D}, \mathcal{M}(\mathcal{D}))\}$ *is* $(\epsilon, \delta)$*-joint differentially private, where* $\Pi_i : \mathcal{D} \rightarrow \mathcal{D}_i$ *is the projection of* $\mathcal{D}$ *onto* $\mathcal{D}_i$.

With the *Billboard Lemma*, we are able to obtain joint differential privacy by first training a differentially private model with data from all tasks, and then finetuning on each task with its local data. We formally introduce our algorithm and corresponding JDP guarantee by using Lemma 1 in Section 4.

Finally, note that our privacy formulation itself is not limited to the multi-task relationship learning framework. For any form of multi-task learning where each task-specific model is obtained by training a combination of global component and local component(e.g. Li et al. (2021)), we can provide a JDP guarantee for the MTL training process by using a differentially private global component.

## 4 PMTL: Private Multi-Task Learning

We now present PMTL, a method for performing joint differentially-private MTL (Section 4.1). We provide both a privacy guarantee (Section 4.2) and utility guarantee (Section 4.3) for our approach.

---

**Algorithm 1** PMTL: Private Mean-Regularized MTL

---

1: **Input:** $m, T, \lambda, \eta, \{w_1^0, \cdots, w_m^0\}, \widetilde{w}^0 = \frac{1}{m}\sum_{k=1}^m w_k^0$
2: **for** $t = 0, \cdots, T-1$ **do**
3:   Global Learner randomly selects a set of tasks $S_t$ and broadcasts the mean weight $\widetilde{w}^t$
4:   **for** $k \in S_t$ in parallel **do**
5:    Each task updates its weight $w_k$ for $E$ iterations, $o_k$ is the last iteration task $k$ is selected
$$w_k^{t+1} = \texttt{ClientUpdate}(w_k^{o_k})$$
6:    Each task sends $g_k^{t+1} = w_k^{t+1} - w_k^t$ back to the global learner.
7:   **end for**
8:   Global Learner computes a noisy aggregator of the weights
$$\widetilde{w}^{t+1} = \widetilde{w}^t + \frac{1}{|S_t|}\sum_{k \in S_t} g_k^{t+1} \boxed{\min\left(1, \frac{\gamma}{\|g_k^{t+1}\|_2}\right)} \boxed{+ \mathcal{N}(0, \sigma^2 \mathbf{I}_{\mathbf{d} \times \mathbf{d}})}$$
9: **end for**
10: **return** $w_1, \cdots, w_m$ as differentially private personalized models

11: $\texttt{ClientUpdate}(\mathbf{w})$
12: **for** $j = 0, \cdots, E-1$ **do**
13:   Task learner performs SGD locally
$$w = w - \eta(\nabla_w l_k(w) + \lambda(w - \widetilde{w}^t))$$
14: **end for**

---

## 4.1 ALGORITHM

We summarize our solver for private multi-task learning in Algorithm 1. Our method is based off of FedAvg (McMahan et al., 2017), a communication-efficient method widely used in federated learning. FedAvg alternates between two steps: (i) each task learner selected at one communication round solves its own local objective by running stochastic gradient descent for $E$ iterations and sending the updated model to the global learner; (ii) the global learner aggregates the local updates and broadcasts the aggregated mean. By performing local updating in this manner, FedAvg has been shown to empirically reduce the total number of communication rounds needed for convergence in federated settings relative to baselines such as mini-batch FedSGD (McMahan et al., 2017). Our private MTL algorithm differs from FedAvg in that: (i) instead of learning a single global model, all task learners collaboratively learn separate, personalized models for each task; (ii) each task learner solves the local objective with the mean-regularization term; (iii) individual model updates are clipped and random Gaussian noise is added to the aggregated model updates to ensure task-level privacy.

To aggregate updates from each task, we assume that we have access to a trusted global learner, i.e., it is safe for some global entity to observe/collect the individual model updates from each task. This is a standard assumption in federated learning, where access to a trusted central server is assumed in order to collect client updates (Kairouz et al., 2019). However, even with this assumption, note that it is possible for any single task learner to infer information about other tasks from the global model, since it is a linear combination of all task specific models and is shared among all task learners.

There are several ways to overcome this privacy risk and thus achieve $(\epsilon, \delta)$-differential privacy. In this paper, we use the Gaussian Mechanism (Dwork & Roth, 2014) during global aggregation as a simple yet effective method, highlighted in the red portion of line 8 in Algorithm 1. In this case, each task learner receives a noisy aggregated global model, making it difficult for any task to leak private information to the others. To apply the Gaussian mechanism, we need to bound the $\ell_2$-sensitivity of each local model update that is communicated to lie in $\mathcal{B} = \{\Delta w | \|\Delta w\|_2 \leq \gamma\}$, as highlighted in the blue part of line 8 in Algorithm 1. Hence, at each communication round, the global learner receives the model updates from each task, and clips the model updates to $\mathcal{B}$ before aggregation. Note that different from DPSGD (Abadi et al., 2016), when we solve the local objective for each selected task at each communication round, our algorithm doesn't clip and perturb the gradient used to update the task-specific model. Instead, since the purpose is to protect task or client-level privacy in multi-task learning, we perform standard SGD locally for each task and only clip and perturb the model update that is sent to the global learner. We formalize the privacy guarantee of Algorithm 1 in Section 4.2.

## 4.2 PRIVACY ANALYSIS

We now rigorously explore the privacy guarantee provided by Algorithm 1. In our optimization scheme, for each task $k$, at the end of each communication round, a shared global model is received. After that the task specific model is updated by optimizing the local objective. We formalize this local task learning process as $h_k : \mathcal{D}_k \times \mathcal{W} \to \mathcal{W}$. Here we simply assume $\mathcal{W} \subset \mathbb{R}^d$ is closed. Define the mechanism for communication round $t$ to be

$$\mathcal{M}^t(\{D_i\}, \{h_i(\cdot)\}, \widetilde{w}^t, \sigma) = \widetilde{w}^t + \frac{1}{|S_t|} \sum_{k \in S_t} h_k(D_k, \widetilde{w}^t) + \beta^t, \tag{6}$$

where $\beta^t \sim \mathcal{N}(0, \sigma^2 \mathrm{I}_{d \times d})$. Note that $\mathcal{M}^t$ characterizes a Sampled Gaussian Mechanism given $\widetilde{w}^t$ as a fixed model rather than the output of a composition of $\mathcal{M}^j$ for $j < t$. To analyze the privacy guarantee of Algorithm 1 over $T$ communication rounds, we define the composition of $\mathcal{M}^1$ to $\mathcal{M}^T$ recursively as $\mathcal{M}^{1:T} = \mathcal{M}^T(\{D_i\}, \{h_i(\cdot)\}, \mathcal{M}^{T-1}, \sigma)$.

**Theorem 1.** *Assume $|S_t| = q$ for all $t$ and the total number of communication rounds is $T$. There exists constants $c_1, c_2$ such that for any $\epsilon < c_1 \frac{q^2}{m^2} T$, the mechanism $\mathcal{M}^{1:T}$ is $(\epsilon, \delta)$-differentially private for any $\delta > 0$ if we choose $\sigma \geq c_2 \frac{\gamma \sqrt{T \log(1/\delta)}}{\epsilon m}$. When $q = m$, $\mathcal{M}^{1:T}$ is $(\epsilon, \delta)$-differentially private if we choose $\sigma = \frac{4\gamma \sqrt{T \log(1/\delta)}}{\epsilon m}$.*

Theorem 1 provides a provable privacy guarantee on the learned global model. When all tasks participate in every communication round, i.e. $q = m$, the global aggregation step in Algorithm 1 reduce to applying Gaussian Mechanism without sampling rather than Sampled Gaussian Mechanism on the average model updates. We provide a detailed proof of Theorem 1 in Appendix A.1.

Note that Theorem 1 doesn't rely on how task learners optimize their local objective. Hence, Theorem 1 is not limited to Algorithm 1 and could be generalized to other local objectives and other global aggregation methods that produce a single model aggregate.

Now we show that Algorithm 1, which outputs $m$ separate models, satisfies joint differential privacy. Given $\widetilde{w}^t$ for any $t \leq T$, we formally define the process that each task learner $k$ optimize its local objective to be $h'_k : \mathcal{D}_k \times \mathcal{W} \to \mathcal{W}$. Note that $h'_k$ is not restricted to be $h_k$ and could represent the optimization process for any local objective. In order to show that Algorithm 1 satisfies JDP, we would apply the *Billboard Lemma* introduced in Section 3. In our case, the average model that is broadcast by the global learner at every communication round is the output of a differentially private learning process. Task learners then individually train their task specific models on the respective private data to obtain personalized models. We now present our main theorem of the JDP guarantee provided by Algorithm 1:

**Theorem 2.** *There exists constants $c_1, c_2$, for any $0 < \epsilon < c_1 \frac{q^2}{m^2} T$ and $\delta > 0$, let $\sigma \geq \frac{c_2 \gamma \sqrt{T \log(1/\delta)}}{\epsilon m}$. Algorithm 1 that outputs $h'_k(D_k, \mathcal{M}^{1:T})$ for each task is $(\epsilon, \delta)$-joint differentially private.*

From Theorem 2, for any fixed $\delta$, the more tasks involved in the learning process, the smaller $\sigma$ we need in order to keep the privacy parameter $\epsilon$ the same. In other words, less noise is required to keep the task-specific data private. When we have infinitely many tasks ($m \to \infty$), we have $\sigma \to 0$, in which case only a negligible amount of noise is needed to add to the model aggregates to make the global model private to all tasks. We provide a detailed proof in Appendix A.1.

**Remark.** Note that privacy guarantee provided by our Theorem 2 is not limited to mean-regularized multi-task learning. For any form of multi-task relationship learning with fixed relationship matrix $\Omega$, as long as we fix the $\ell_2$-sensitivity of model updates and the noise scale of the Gaussian mechanism applied to the statistics broadcast to all task learners, the privacy guarantee induced by this aggregation step is fixed, regardless of the local objective being optimized. For example, as a natural extension of our mean-regularized MTL objective, consider the case where task learners are partitioned into fixed clusters and optimize the mean-regularized MTL objective within each cluster, as in Evgeniou et al. (2005). In this scenario, Theorem 2 directly applies to the algorithm run on each cluster.

## 4.3 CONVERGENCE ANALYSIS

As discussed in Section 3, we are interested in the following task-specific objective:

$$f_k(w_k; \widetilde{w}) = l_k(w_k) + \frac{\lambda}{2} \|w_k - \widetilde{w}\|_2^2 \tag{7}$$

where $\widetilde{w}$ is an estimate for the average model $\overline{w}$; $l_k(w_k)$ is the empirical loss for task $k$; $w_k \in \mathbb{R}^d$.

Here, we analyze the convergence behavior in the setting where a set $S_t$ of $q$ tasks participate in the optimization process at every communication round. Further, we assume the number of local optimization steps $E = 1$. We present the following convergence result:

**Theorem 3** (Convergence under nonconvex loss). *Let $f_k$ be $(L + \lambda)$-smooth. Assume $\gamma$ is sufficiently large such that $\gamma \geq \max_{k,t} \|\nabla_{w_k^t} f_k(w_k^t; \widetilde{w}^t)\|_2$. Further let $f_k^* = \min_{w,\bar{w}} f_k(w; \bar{w})$ and $p = \frac{q}{m}$. If we use a fixed learning rate $\eta_t = \eta = \frac{1}{pL + (p - \frac{1}{p})\lambda}$, Algorithm 1 satisfies:*

$$\frac{1}{mT} \sum_{t=0}^{T-1} \sum_{k=1}^{m} \|\nabla f_k(w_k^t; \widetilde{w}^t)\|^2 \leq \mathcal{O}\left(\frac{1}{mT}\right) + \frac{\mathcal{O}\left(L + \lambda + \frac{\lambda}{p^2}\right) \sum_{t=0}^{T-1} B_{t+1}}{T} + \mathcal{O}\left(d\sigma^2\right). \quad (8)$$

*where*

$$B_t = \max_k f_k(w_k^t; \widetilde{w}^t). \quad (9)$$

*Let $\sigma$ chosen as we set in Theorem 2. Take $T = \mathcal{O}\left(\frac{m}{\lambda d\gamma^2}\right)$, the right hand side is bounded by*

$$\frac{\lambda d\gamma^2}{m^2} \sum_{t=0}^{T-1} \sum_{k=1}^{m} \|\nabla f_k(w_k^t; \widetilde{w}^t)\|^2 \leq \mathcal{O}\left(\frac{d\gamma^2}{m^2}\right) + \mathcal{O}\left(d\gamma^2\right) \sum_{t=0}^{T-1} B_{t+1} + \mathcal{O}\left(\frac{1}{m}\right) \frac{\log(1/\delta)}{\epsilon^2}. \quad (10)$$

We provide a formal statement and full proof of Theorem 3 in Appendix A.2. The upper bound in (30) consists of two parts: error induced by the gradient descent algorithm and error induced by the Gaussian Mechanism. When $\sigma = 0$, Algorithm 1 recovers a non-private mean-regularized multi-task learning solver.

**Corollary 4.** *When $\sigma = 0$, Algorithm 1 with $(L + \lambda)$-smooth and nonconvex $f_k$ satisfies*

$$\frac{1}{mT} \sum_{t=0}^{T-1} \sum_{k=1}^{m} \|\nabla f_k(w_k^t; \widetilde{w}^t)\|^2 \leq \mathcal{O}\left(\frac{1}{mT}\right) + \frac{\mathcal{O}\left(L + \lambda + \frac{\lambda}{p^2}\right) \sum_{t=0}^{T-1} B_{t+1}}{T}. \quad (11)$$

By Theorem 2, given fixed $\epsilon$, $\sigma^2$ grows linearly with respect to $T$. Hence, given the same privacy guarantee, larger noise is required if the algorithm is run for more communication rounds. Note that in Theorem 3, the upper bound consists of $\mathcal{O}(\frac{1}{m\epsilon^2})$, which means when there are more tasks, the upper bound becomes smaller while the privacy parameter remains the same. On the other hand, Theorem 3 also shows a privacy-utility tradeoff using our Algorithm 1: the upper bound grows inversely proportional to the privacy parameter $\epsilon$. We also provide a convergence analysis of Algorithm 1 with strongly-convex losses in Theorem 5 below (formal statement and proof in Appendix A.3).

**Theorem 5** (Convergence under strongly-convex loss). *Let $f_k$ be $(L + \lambda)$-smooth and $(\mu + \lambda)$-strongly convex. Assume $\gamma$ is sufficiently large such that $\gamma \geq \max_{k,t} \|\nabla_{w_k^t} f_k(w_k^t; \widetilde{w}^t)\|_2$. Further let $w_k^* = \arg\min_w f_k(w; \bar{w}^*)$, where $\bar{w}^* = \frac{1}{m} \sum_{k=1}^{m} w_k^*$ and $p = \frac{q}{m}$. If we use a fixed learning rate $\eta_t = \eta = \frac{c}{\frac{L-2}{p} + \lambda p}$ for some constant $c$ such that $0 \leq \eta p(c-2)(\mu + \lambda) \leq 1$, Algorithm 1 satisfies:*

$$\frac{1}{m} \Delta_T \leq (1 - \eta p(c-2)(\mu + \lambda))^T \left(\frac{1}{m} \Delta_0 - C\right) + C, \quad (12)$$

*where $\Delta_t = \sum_{k=1}^{m} f_k(w_k^t; \widetilde{w}^t) - f_k(w_k^*; \widetilde{w}^*)$, $C = \mathcal{O}\left(\frac{\left(\sqrt{d}\sigma + \sqrt{\frac{2}{\lambda}B}\right)^2}{\eta}\right)$, $B = \max_t \max_k f_k(w_k^t; \widetilde{w}^t)$.*

*Let $\sigma$ be chosen as in Theorem 2, then there exists $T = \mathcal{O}\left(\frac{m(c^2 - 2c)(\mu + \lambda)}{\lambda\left(\frac{L-2}{p^2} + \lambda\right)d\gamma^2}\right)$ such that*

$$\frac{1}{m} \Delta_T \leq (1 - \eta p(c-2)(\mu + \lambda))^T \left(\frac{1}{m} \Delta_0 - \frac{\log(1/\delta)}{m\epsilon^2} - \mathcal{O}\left(\frac{B}{\eta}\right)\right) + \frac{\log(1/\delta)}{m\epsilon^2} + \mathcal{O}\left(\frac{B}{\eta}\right). \quad (13)$$

As with Corollary 4, we recover the bound of the non-private mean-regularized MTL solver for $\sigma=0$.

**Corollary 6.** *When $\sigma = 0$, Algorithm 1 with $(L+\lambda)$-smooth and $(\mu+\lambda)$-strongly convex $f_k$ satisfies*

$$\frac{1}{m}\Delta_T \leq (1 - \eta p(c-2)(\mu+\lambda))^T \left(\frac{1}{m}\Delta_0 - \frac{B}{\eta p(c-2)(\mu+\lambda)}\right) + \frac{B}{\eta p(c-2)(\mu+\lambda)}. \quad (14)$$

## 5 EXPERIMENTS

In this section, we empirically evaluate our private MTL solver on several federated learning benchmarks. Specifically, we demonstrate the privacy-utility trade-off of training a MTL objective compared with training a global model. We then compare results of performing local finetuning after learning an MTL objective with local finetuning after learning a global objective.

### 5.1 SETUP

For all experiments, we evaluate the test accuracy and privacy parameter of our private MTL solver given a fixed clipping bound $\gamma$, variance of Gaussian noise $\sigma^2$, and communication rounds $T$. All experiments are performed on common federated learning benchmarks as a natural application of multi-task learning. We provide a detailed description of datasets and models in Appendix A.4. Each dataset is naturally partitioned among $m$ different clients. Under such a scenario, each client can be viewed as a task and the data that a client generates is only visible to the local task learner.

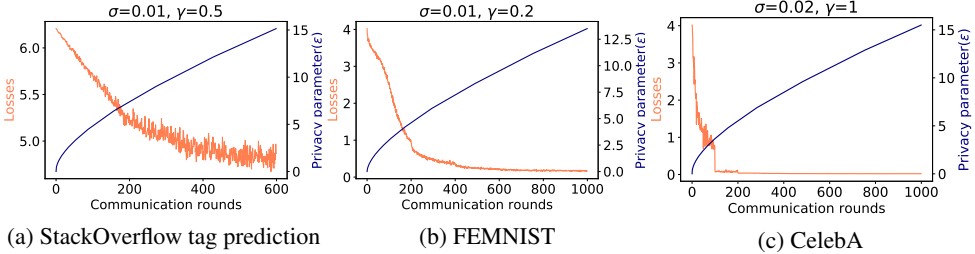

|  |  |  |
|---|---|---|
| (a) StackOverflow tag prediction | (b) FEMNIST | (c) CelebA |

Figure 2: Loss and privacy parameter vs. communication rounds for PMTL. The blue line shows the change of privacy parameter $\epsilon$ in terms of number of communication rounds during training. The orange line shows the average training loss across all tasks.

### 5.2 PRIVACY-UTILITY TRADE-OFF OF PMTL

We first explore the training loss (**orange**) and privacy parameter $\epsilon$ (**blue**) as a function of communication rounds across three datasets (Figure 2). Specifically, we evaluate the average loss for all the tasks and $\epsilon$ given a fixed $\delta$ after each round, where $\delta$ is set to be $\frac{1}{m}$ for all experiments. In general, for a fixed clipping bound $\gamma$ and $\sigma$, we see that the method converges fairly quickly with respect to the resulting privacy, but that privacy guarantees may be sacrificed in order to achieve very small losses.

To put these results in context, we also compare the test performance of our private MTL solver with that of training a global model. In particular, we use FedAvg (McMahan et al., 2017) to train a global model. At each communication round, task learners solve their local objective individually. Assuming the global learner is trustworthy, while aggregating the model updates from all tasks, the global learner applies a Gaussian Mechanism and sends the noisy aggregation back to the task learners. As a result, private FedAvg differs from our private MTL solver in the following two places: (i) the MTL objective solved locally by each task learner has an additional mean-regularized term; (ii) the MTL method evaluates on one task-specific model for every task while the global method evaluates all tasks on one global model. For each dataset, we select privacy parameter $\epsilon \in [0.05, 0.1, 0.2, 0.4, 0.8, 1.6, 2.0, 4.0]$. For each $\epsilon$, we select the $\gamma$, $\sigma$, and $T$ that result in the best validation accuracy for a given $\epsilon$ and record the test accuracy. A detailed description of hyperparameters is listed in Appendix A.5. We plot the test accuracy with respect to the highest validation accuracy given one $\epsilon$ for both private MTL model and private global model. The results are shown in Figure 3.

In all three datasets, our private MTL solver achieves higher test accuracy compared with training a private global model with FedAvg given the same $\epsilon$. Moreover, the proposed mean regularized MTL solver is able to retain an advantage over global model even with noisy aggregation. In particular, for small $\epsilon < 1$, adding random Gaussian noise during global aggregation amplifies the test accuracy

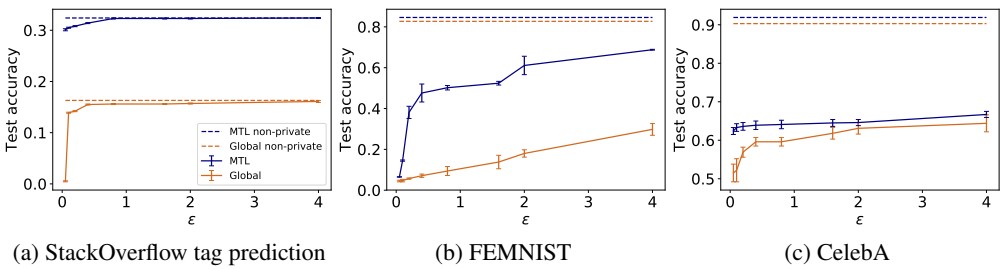

Figure 3: Comparison of PMTL and training a private global model.

difference between our MTL solver and FedAvg. Under the StackOverflow task, both methods obtain test accuracy close to the non private baseline for large $\epsilon$. To demonstrate that applying private MTL has an advantage over private global training more generally, we also compared our PMTL method with private FedProx (Li et al., 2020b). The results (which mirror Figure 3) are in Appendix A.6.

Table 1: Comparison of private MTL and private Global model with different local finetuning methods. $\epsilon = \infty$ corresponds to the case where no noise and clipping happened, i.e., training non-privately. The higher accuracy between MTL and Global given the same $\epsilon$ and finetuning method is **bolded**.

| FEMNIST | $\epsilon = 0.1$ | | $\epsilon = 0.8$ | | $\epsilon = 2.0$ | | $\epsilon = \infty$ | |
|---|---|---|---|---|---|---|---|---|
| | MTL | Global | MTL | Global | MTL | Global | MTL | Global |
| Vanilla Finetuning | **0.645 ± 0.013** | 0.606 ± 0.017 | 0.640 ± 0.016 | **0.648 ± 0.017** | **0.677 ± 0.008** | 0.653 ± 0.010 | **0.832 ± 0.005** | 0.812 ± 0.009 |
| Mean-regularization | **0.608 ± 0.011** | 0.581 ± 0.011 | **0.605 ± 0.008** | 0.574 ± 0.006 | **0.656 ± 0.009** | 0.633 ± 0.003 | 0.826 ± 0.011 | **0.839 ± 0.006** |
| Symmetrized KL | **0.486 ± 0.012** | 0.348 ± 0.005 | **0.584 ± 0.012** | 0.481 ± 0.016 | **0.662 ± 0.016** | 0.565 ± 0.019 | **0.839 ± 0.006** | 0.829 ± 0.015 |
| EWC | **0.663 ± 0.002** | 0.556 ± 0.001 | 0.595 ± 0.004 | **0.607 ± 0.007** | **0.681 ± 0.002** | 0.666 ± 0.001 | **0.837 ± 0.001** | 0.823 ± 0.005 |

## 5.3 PMTL WITH LOCAL FINETUNING

Finally, in federated learning, previous works have shown local finetuning with different objectives is helpful for improving utility while training a differentially private global model (Yu et al., 2020). In this section, after obtaining a private global model, we explore locally finetuning the task specific models by optimizing different local objective functions. In particular, we use common objectives which (i) naively optimize the local empirical risk (Vanilla Finetuning), or (ii) encourage minimizing the distance between local and global model under different distance metrics (Mean-regularization, Symmetrized KL, EWC (Kirkpatrick et al., 2017; Yu et al., 2020)). The results are listed in Table 1. When $\epsilon = \infty$ (the non-private setting), global with mean-regularization finetuning outperforms all MTL+finetuning methods. However, when we add privacy to both methods, private MTL+finetuning has an advantage over global with finetuning on different finetuning objectives. In some cases, e.g. using Symmetrized KL as the finetuning objective, the test accuracy gap between private MTL with fintuning and private global with finetuning is amplified when $\epsilon$ is small compared to the case where no privacy is added during training.

## 6 CONCLUSION AND FUTURE WORK

In this work, we define notions of task-level privacy for multi-task learning and propose a simple method for differentially private mean-regularized MTL. Theoretically, we provide both privacy and utility guarantees for our approach. Empirically, we show that private mean-regularized MTL retains advantages over training a private global on common federated learning benchmarks. In future work, we are interested in building on our results to explore privacy for more general forms of MTL, e.g., the family of objectives in (1) with arbitrary matrix $\Omega$. We are also interested in studying how task-level privacy relates to algorithmic fairness in the MTL setting.

**Ethics Statement.** In this work we aim to raise awareness about privacy issues that potentially exist when performing multi-task learning, which is commonly used to model heterogeneous data in areas such as healthcare, finance, and IoT computing. We are particularly motivated by the application of cross-device federated learning, where numerous forms of multi-task learning have been proposed to improve accuracy, but the privacy implications remain unclear. To ensure common notions of task or client-level privacy in this setting while retaining the demonstrated accuracy benefits of MTL, we have provided formal privacy definitions for multi-task learning. Building on these definitions, we proposed a practical algorithm and studied its performance both theoretically and empirically for the special case of mean-regularized multi-task learning objective. We hope this work will be a starting point for future extensions that study the issue of privacy in multi-task learning.

**Reproducibility Statement.** To enable reproducibility in our theoretical and empirical results, we provide:

- Proofs of Theorem 1, 2, 3, 5 in Appendix A.1, A.2, A.3.
- Dataset and model details in Appendix A.4.
- Hyperparameters for tuning in Appendix A.5.
- Code used for running experiments in Supplementary Material.

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
