# OpenReview forum: "Private Multi-Task Learning: Formulation and Applications to Federated Learning"
_ICLR.cc/2022/Conference — ICLR 2022 Submitted_

### Official Review · Reviewer_T1p3 · 2021-11-02

**Correctness:** 3
**Technical Novelty And Significance:** 2
**Empirical Novelty And Significance:** 3
**Recommendation:** 3
**Confidence:** 3

**Main Review:**

Overall, I find Sections 1-3 to be strong: the paper is well-written and the problem is well-formulated and nicely described. My main problem is with the convergence results, which are difficult to interpret due to too many parameters and no informal bound is provided. Additionally, it is not clear that the bounds are even non-trivial (see below for more discussion). My other main point of confusion is that the authors say they consider JDP as a relaxation of TLDP to provide better utility, but then their algorithm actually appears to provide the stronger (by Billboard Lemma) notion of TLDP (due to Theorem 1); so what is the role of JDP in their results?

Detailed Review:
p.1:

is task-level DP essentially the same as user-level DP/client-level DP (which is considered in McMahan et al 2018)? If so, state that--these terms are more familiar to the DP community than task-level DP.

emphasize "as long as client k keeps their data private"

p.3

Motivate Eq (1). Goal is to train a model that has small average loss over all tasks, right? So why use a regularizer?

Define $\ell_k$ and $x_i$

Missing parentheses in Eq (2)

p. 4

clarify that strong guarantee of JDP hinges on each client keeping its own model private

I believe Kearns et al 2014 is wrong citation for Lemma 1. Shouldn't it be Hsu et al?

Lemma 1: $D_k$ ->  $D_i$

p.6

Theorem 1: missing restriction on $\epsilon$. Proof of Abadi et al (2016) Theorem 1 shows that some restriction (roughly $\epsilon \leq \ln(1/\delta)$) is needed; also, you should refer to what you provide as TLDP or user-level/client-level DP (in Theorem 1 and in Definition 1)--DP is too vague, as there are many notions of DP

**Why study JDP if your algorithm satisfies the stronger notion of TLDP?**

p.7

If you assume $\gamma \geq \| \nabla f_k(w) \|$ (i.e. loss is effectively Lipschitz), then it seems you do not need to clip the gradients (and clipping will never actually occur in the algorithm)--am I right?

Why assume loss is $L + \lambda$-smooth and not $L$ smooth?

Should replace min by inf unless you are assuming compactness of domain; are you? I don't the domain was formally defined. I don't remember seeing $d$ defined either.

**Convergence bounds: if $f_k \leq 1$ so $B_t = 1$, then the second term in Eq (11) becomes essentially trivial $O(L + \lambda + ...)$** And that is the non-private case, so the DP bounds can only get "worse".

Also, **the bounds in (8)-(10) and (12) are too complicated: should provide simplified version with dependence on key parameters m, d, epsilon, delta displayed**; and even the formal version should not be simplified to include only the key parameters as well as the smoothness and boundedness parameters, and non-dominant terms should be omitted.




**Summary Of The Paper:**

The paper considers the problem of multi-task learning  (MTL) with task-level differential privacy (TLDP) constraints. The authors propose using joint differential privacy (JDP) for MTL and provide a JDP MTL algorithm, which is a DP variation of FedAvg. They prove the privacy of their algorithm and provide convergence results for smooth nonconvex and strongly convex losses. They then provide numerical experiments.

**Summary Of The Review:**

The paper considers/formulates an interesting problem and is generally well-written. Unfortunately, the convergence results do not appear to be strong (or clear). Additionally, there is confusion about the notion of privacy that their algorithm provides and the role that JDP is playing in the paper. For these reasons I cannot recommend acceptance for the current form of the paper.

---

> ### Author Response · Authors · 2021-11-17
> **Response T1p3**
>
> We thank the reviewer for taking the time to review our work and for your detailed feedback. We discuss the points raised below:
> - **Role of JDP**: Our work aims to provide a formal privacy guarantee for multi-task learning (MTL), where a separate model is learned for every task learner (commonly referred to as a client in FL). To do so, we developed Algorithm 1, which satisfies task-level joint differential privacy, a relaxed notion of differential privacy on the task/client level, for the multi-task learning objective. It’s worth noting that our algorithm only provides traditional TLDP when learning the global model but not for learning $m$ personalized models. Specifically, the TLDP guarantee for learning the global model results from noisy global aggregation (Theorem 1) while the JDP guarantee for learning multiple models results from applying the Billboard Lemma to a TLDP process (Theorem 2). Hence, the use of Billboard Lemma does not result in TLDP itself.  As discussed in the common response, the notion of traditional, task-level DP is incompatible with MTL, which is why it is necessary to use JDP.
> - **Client-level vs. task-level**: We use the term ‘task-level’ privacy since this has been formalized by prior work for the related problem of differentially private meta-learning [1].  While we focus on federated learning, our approaches are applicable more generally to any scenario where it is desirable to use multi-task learning while ensuring the privacy of each underlying task. However, you are correct that when referring to task-level privacy in the context of federated learning, this can equivalently be viewed as ‘client-level’ privacy. We state this at multiple points in the paper, but have also clarified it further by adding a discussion directly in the introduction. Thanks for this suggestion!
> - **Goal of MTL**: The goal in MTL (as in general ML) is to find model(s) that generalize well to new data. In multi-task learning, the idea is that for a given task (e.g., a client in FL), using the information from related tasks (other clients) can act as a form of inductive bias to improve generalization. Mean-regularized MTL (proposed by Evgenio & Pontil in 2004 [1]) is one example of this where you solve each task independently but ensure that the task-specific models do not deviate too much from the average model. This simple form of MTL has been shown through numerous examples (including many applications in federated learning [3-6]) to improve generalization performance. It is well-known to be useful for the field of FL, where each client’s task may differ slightly due differences in the underlying data distributions (i.e., the data is non-identically distributed).
> - **Definitions**: x_i is the data and lk(wk) in eq (7) is the empirical loss. We have clarified this in our revised PDF. Thanks!
> - **JDP citation**: Thank you for catching this! The correct citation for Lemma 1 is Hsu et al. We have corrected this in our revision, along with correcting $D_k$ to be $D_i$.
> - **Smoothness assumption**: We assume the empirical loss $l_k$ to be $L$-smooth. As a result, after adding the regularization term, the entire objective is $L+\lambda$-smooth.
> - **Definition of d**: $d$ is the dimension of the model parameter, i.e. $w\in\mathbb{R}^d$. We have added this in our revision.
> - **Theorem 1**: We believe you may have missed that in our Theorem 1, we have made the assumption that $\epsilon<c_1\frac{q^2}{m^2}T$, which is similar to the assumption made in Theorem 1 in Abadi et al.
> - **Clipping in Theorem 3**: The study of clipping in SGD is complex and warrants a separate line of work [7,8,9]. In this work, as you point out, we instead assume the Lipschitzness of the loss function, which is standard in prior convergence analyses of private SGD [10,11]. Specifically, in Theorem 3, we have assumed that $\gamma\geq\max_{k,t}\|\nabla_{w_k^t}f_k(w_k^t;\tilde{w}^t)\|_2$.
> - **Convergence bounds**: This extra source of ‘error’ stems from the mean-regularization term. In the non-private scenario, where the first constant term vanishes, the local objective can only converge to a neighborhood of the optimum. If this were not the case, the regularization term would asymptotically go to 0 as the number of communication rounds increases, in which case every personalized model would just become the average model, losing the purpose of using MTL to train personalized models. This result is therefore expected/standard, and is in line with other previous works that study the same objective but with different solvers [e.g., 6].
> - **Simplified bounds**: Thanks for suggesting simplified versions of the bounds. We have presented the informal bound for both nonconvex and convex convergence analysis in the revised version, and have moved the full bounds to the appendix.

---

> > ### Author Response · Authors · 2021-11-17
> > **Response T1p3 (continued)**
> >
> > [1] Li, J., Khodak, M., Caldas, S., & Talwalkar, A. Differentially Private Meta-Learning, ICLR 2020.
> >
> > [2] T. Evgeniou, M. Pontil. Regularized multi-task learning. KDD, 2004
> >
> > [3] V. Smith, C. Chiang, M. Sanjabi, and A. Talwalkar. Federated multi-task learning. NeurIPS, 2017.
> >
> > [4] F. Hanzely, S. Hanzely, S. Horvath, and P. Richtarik. Lower bounds and optimal algorithms for personalized federated learning. NeurIPS, 2020.
> >
> > [5] C. Dinh, N. Tran, and T. Nguyen. Personalized federated learning with moreau envelopes. NeurIPS, 2020.
> >
> > [6] F. Hanzely and P. Richtarik. Federated learning of a mixture of global and local models. arXiv preprint arXiv:2002.05516, 2020.
> >
> > [7] Chen, X., Wu, S. Z., & Hong, M. Understanding gradient clipping in private SGD: a geometric perspective. Advances in Neural Information Processing Systems, 33, 2020.
> >
> > [8] Song, S., Steinke, T., Thakkar, O., & Thakurta, A. Evading Curse of Dimensionality in Unconstrained Private GLMs via Private Gradient Descent. arXiv preprint arXiv:2006.06783, 2020.
> >
> > [9] Zhang, X., Chen, X., Hong, M., Wu, Z. S., & Yi, J. Understanding Clipping for Federated Learning: Convergence and Client-Level Differential Privacy. arXiv preprint arXiv:2106.13673, 2021.
> >
> > [10] Bassily, R., Smith, A., & Thakurta, A. (2014, October). Private empirical risk minimization: Efficient algorithms and tight error bounds. In 2014 IEEE 55th Annual Symposium on Foundations of Computer Science (pp. 464-473). IEEE.
> >
> > [11] Di Wang and Jinhui Xu. Differentially private empirical risk minimization with smooth nonconvex loss functions: A non-stationary view. In Proceedings of the AAAI Conference on Artificial Intelligence, volume 33, pages 1182–1189, 2019.

---

> > > ### Comment · Reviewer_T1p3 · 2021-11-18
> > > **Still concerned about convergence, and some other things**
> > >
> > > Thank you for the detailed response and revised version. My main concern is still the convergence rate. Even ignoring the privacy term and  assuming p = 1 (best case scenario) and smoothness is 1, then looking at Eq. 11, the convergence rate is O(1) if f is bounded by 1. In the homogeneous case, this is no better than the trivial algorithm that returns 0  (which is also differentially private). I did take a brief look at [6] referenced above and all of their rates appeared to be non-trivial (e.g. in their theorem 4.5, even accounting for heterogeneity in tasks, they scaled as 1/n). Please correct me if I misunderstand something, but to me this seems like a big problem with the stated results in Theorem 3.
> > >
> > > Another concern is that the units of different quantities in Theorem 3 (and possibly other results too, but I only checked Theorem 3) do not appear to match. For example, T should be unit-free, but gamma has units F/w (as it measures gradient) and lambda has units F/w^2 (as it measures gradient changes i.e. smoothness), so m/lambda d gamma^2 has units w^4/F^3 which does not make sense. Units should be carefully checked throughout.
> > >
> > > Also, the informal result can be further simplified. For example, p \leq 1 always so lambda/p dominates lambda.
> > >
> > > __
> > >
> > > If clipping is not actually used for informal results (because Lipschitzness is assumed), then, in my opinion, the algorithm presented should be simplified (no clipping), so it corresponds to the algorithm analyzed in deriving convergence bounds. If you use it in the experiments, you can add a note there.

---

> > > > ### Author Response · Authors · 2021-11-20
> > > > **Response**
> > > >
> > > > Thank you for your quick reply. We respond to the follow-up comments below.
> > > >
> > > > **[Convergence rate]**: We wish to clarify two aspects regarding Equation (11) [ignoring privacy terms]. First, given the nature of the mean-regularized MTL objective, it is expected that the local objective will only converge to within a neighborhood of the optimal. Consider the following simple mean-estimation problem as an example. Assume that we have $m$ different clients/tasks, each with local data $x_i$ and local model $w_i$. The mean-regularized MTL objective for this problem would be $\frac{1}{m}\sum_{i}(x_i-w_i)^2+\frac{1}{m}\sum_i(w_i-\bar{w})^2$, which is greater than $\frac{1}{2m}\sum_i(x_i-w_i+w_i-\bar{w})^2 = \frac{1}{2m}\sum_i(x_i-\bar{w})^2$.  Note that this lower bound neither converges to 0 as $\bar{w}$ changes over time nor diminishes with increasing $m$. Therefore, it is not surprising that we do not see $\mathcal{O}(\frac{1}{m})$ appear in the constant term in our convergence bound.
> > > > Second, we wish to clarify that the bound in Theorem 4.5 of [6] does not appear to scale as $\mathcal{O}(\frac{1}{n})$. Assume the regularization term is uniformly bounded, i.e. $\|x_i(\lambda)-\bar{x}(\lambda)\|^2\leq B$. Then the term $\frac{2n\alpha\sigma^2}{\mu}$ will be roughly $\mathcal{O}(\alpha B)$ (not $\mathcal{O}(\frac{1}{n})$), since $B$ will be summed $n$ times in the calculation of $\sigma^2$.
> > > >
> > > > **[Units]**: We are a bit confused about what the reviewer means here by ‘units’. Specifically, could you explain what you mean by $F/w$ and $F/w^2$? In our setting, $w$ is a vector and $F$ is the objective value. $\gamma$ and $\lambda$ are scalar values. We are unsure what you mean when saying that $\gamma$ (which is a scalar) has the unit of $F/w$, which is a vector.
> > > >
> > > > **[More simplified bound]**: You are correct that $\lambda$ is dominated by $\lambda/p^2$. We chose to include the $\lambda$ term in order to emphasize that $f_k$ is $(L+\lambda)$-smooth. However we are happy to remove the additional $\lambda$ term to further simplify the informal bound.
> > > >
> > > > **[Clipping]**: We have included clipping in Algorithm 1 as we want to present an algorithm that always satisfies differential privacy even in the case that assumptions like Lipschitzness do not hold. Performing clipping on gradients/model updates is common in practice and we use this strategy in our empirical study (see Figure 4,5,6). This form of clipping is similarly performed in Abadi et al. [1] and many follow-up works [e.g., 2-4]. Regarding analyses, almost all existing convergence analyses on DP-SGD and related algorithms do not analyze the effects of clipping and instead rely on assumptions like Lipschitzness of the loss functions [5,6]. Understanding the effects of clipping is not the focus of our paper and requires a separate line of recent work (see, e.g., [2-4]).
> > > >
> > > > [1] Abadi, M., Chu, A., Goodfellow, I., McMahan, H. B., Mironov, I., Talwar, K., & Zhang, L. Deep learning with differential privacy. In Proceedings of the 2016 ACM SIGSAC conference on computer and communications security, 2016.
> > > >
> > > > [2] Chen, X., Wu, S. Z., & Hong, M. Understanding gradient clipping in private SGD: a geometric perspective. Advances in Neural Information Processing Systems, 33, 2020.
> > > >
> > > > [3] Song, S., Steinke, T., Thakkar, O., & Thakurta, A. Evading Curse of Dimensionality in Unconstrained Private GLMs via Private Gradient Descent. arXiv preprint arXiv:2006.06783, 2020.
> > > >
> > > > [4] Zhang, X., Chen, X., Hong, M., Wu, Z. S., & Yi, J. Understanding Clipping for Federated Learning: Convergence and Client-Level Differential Privacy. arXiv preprint arXiv:2106.13673, 2021.
> > > >
> > > > [5] Kairouz, P., Diaz, M.R., Rush, K. &amp; Thakurta, A.. (2021). (Nearly) Dimension Independent Private ERM with AdaGrad Rates via Publicly Estimated Subspaces. Proceedings of Thirty Fourth Conference on Learning Theory.
> > > >
> > > > [6] Asi, H., Duchi, J., Fallah, A., Javidbakht, O., & Talwar, K. Private Adaptive Gradient Methods for Convex Optimization. In International Conference on Machine Learning (pp. 383-392), 2021.

---

> > > > > ### Author Response · Authors · 2021-12-05
> > > > > **Response (follow up)**
> > > > >
> > > > > Dear Reviewer:
> > > > >
> > > > > Thanks again for your detailed comments and suggestions. We would like to check to see whether our response has adequately resolved your concerns? If not, we would appreciate it if you could let us know whether there are any further concerns that we can discuss. As mentioned, the form of our bounds is expected given the MTL objective we aim to solve, and our results are also in line with existing work studying this objective. We are not sure what is meant by 'units' in your response and would be grateful if you could clarify this point.

---

### Official Review · Reviewer_C2cr · 2021-11-02

**Correctness:** 3
**Technical Novelty And Significance:** 3
**Empirical Novelty And Significance:** Not applicable
**Recommendation:** 6
**Confidence:** 2

**Main Review:**

They give a task-level private algorithm for multi-task learning, which is very important in machine learning, and show the theoretical convergence analysis. The structure is clear and writing is good. The experiment results can also support their theory.
Typo. Section 5.3 “common” rather than “commons”


**Summary Of The Paper:**

Many problems in machine learning rely on multi-task learning (MTL), in which the goal is to solve multiple related machine learning tasks simultaneously. In this work, authors formalize notions of task-level privacy for MTL via joint differential privacy (JDP). They propose an algorithm for mean-regularized MTL, an objective commonly used for applications in personalized federated learning, subject to JDP. Then analyze objective and solver, providing certifiable guarantees on both privacy and utility. The method allows for improved privacy/utility trade-offs relative to global baselines across common federated learning benchmarks.

**Summary Of The Review:**

The work is complement, although only one algorithm for one problem, the analysis and experiments are sufficient.

---

> ### Author Response · Authors · 2021-11-17
> **Response C2cr:**
>
> Thank you for your comments and your positive feedback for this work. We appreciate your time reviewing it! We have fixed the typo you identified in our revised PDF.

---

### Official Review · Reviewer_fBjV · 2021-11-04

**Correctness:** 4
**Technical Novelty And Significance:** 2
**Empirical Novelty And Significance:** 2
**Recommendation:** 3
**Confidence:** 4

**Main Review:**

Strengths:
This paper proposes a new idea of using a relaxation of differential privacy in the multitask learning framework.
It motivates work in the direction of looking at application specific relaxations of differential privacy, to give more realistic algorithms which perform better than the ones which satisfy the strong differential privacy constraint.

Weaknesses/Questions:


1. Is $x_i$ the data? Is $l_k(w_k)$ in eq. (7) the population loss or the empirical loss?

2. Some more results like comparison of rates to algorithms that are not only joint differentially private, but differentially private would be useful. Are the improvements only in constant factors or are the improvements in terms of dimension dependent or number of samples dependent terms?

3. Can the authors give some sense of what are the best rates you can under joint differential privacy and how does that compare with the given algorithm?

4. Is there any motivation as to why the number of local optimization steps was chosen to be 1? In more realistic scenarios, each task would take multiple descent steps to personalize better, right?

5. Can you show a short calculation to ensure the constant multiplying the exponentially decaying term is always positive?

6. The last two terms in equation (13) seem fishy, does it mean the error doesn’t decay by increasing the number of iterations?

7. The values of epsilon in figure 2, seem very high for the loss to decay enough.

8. It’s not surprising that private MTL does better than the private global models, by the sheer number of parameters that private MTL allows (it’s ~m times the private global model parameters). A more fair comparison would be to compare against an algorithm that is globally differentially private instead of joint differentially private.


**Summary Of The Paper:**

In this paper, the authors formalize a notion of task-level differential privacy using joint differential privacy, a concept known in the differentially private mechanism design literature. They formulate the problem of multitask learning under differential privacy under this relaxed definition. The motivation is that there is no need for privacy of the data of a particular task from the task itself, but since you are sharing information in multitask learning, you want to protect against other tasks learning about your private information. They provide an algorithm with convergence guarantees to do multitask learning in this framework and perform some experiments to test this algorithm in the application of federated learning

**Summary Of The Review:**

The idea to use joint differential privacy instead of differential privacy in MTL is interesting. The privacy analysis and the convergence rates presented are fairly standard. The theory doesn’t seem to stand on it’s own and more experiments can be done to rigorously test the introduction of this idea. Hence, I believe in it’s current form the paper is not ready for publication.

---

> ### Author Response · Authors · 2021-11-17
> **Response fBjV**
>
> We thank the reviewer for taking time to review our work and for your detailed comments. We provide answers to the points that were raised below:
>
>
> - **Clarifications**: x_i is the data, and lk(wk) in eq (7) is the empirical loss. We have defined these terms in our revised PDF.
> - **Comparisons with traditional DP methods**: As mentioned in the common response above, the notion of traditional, task-level DP is incompatible with MTL, which is why it does not make sense to consider MTL+DP as a baseline. Since traditional DP is not an option, we instead propose JDP as an appropriate formalization for protecting task-level privacy of an MTL algorithm. Although MTL is a popular approach in federated learning, we are unaware of any work that has formalized notions of task-level DP for MTL. In terms of comparisons, we compare our private MTL method to the natural baseline of training a *global model* via DP in Section 5.2 (Figure 3).
> - **JDP Rates**: Previous work has shown that in the private matchings and allocations problem [1], it is impossible to achieve non-trivial accuracy under standard DP; instead, a strong accuracy guarantee could be provided under JDP. However, in the context of SGD, previous analyses are not directly applicable, and we are not aware of any prior work that formally studied the best rates of jointly differential private SGD. However, as a sanity check, our convergence rates are similar to existing rates of differentially private SGD [2].
> - **Local steps**: Although the number of steps for local optimization has the potential to increase personalization, the level of personalization is more directly controlled by the value of $\lambda$ in the mean-regularized MTL formulation (see Equation 7). For example, as $\lambda$ goes to 0, the objective reduces to solving separate local models; as $\lambda$ goes to $\infty$, the objective reduces to solving a global model. In light of this, we instead focus on the impact of $\lambda$ and set the number of local steps to be 1 as it simplifies the analysis and makes the final bound easier to parse. However, it would be easy to extend this to consider multiple local steps (you could interpret this in our current analysis as using a smaller value of $\lambda$).
> - **Equation 12**: We assume that the reviewer is referring to Equation (12). The minimum value for $\Delta_0$ could be 0. In that case, the right hand side for Equation (12) becomes $\frac{m\lambda\left(d\sigma^2+2\sqrt{d}\sigma\sqrt{\frac{2}{\lambda}B}+\frac{1}{\lambda}B\right)}{\eta p(c-2)(\mu+\lambda)} (1-(1-\eta p(c-2)(\mu+\lambda))^T)$. Since we have restricted the choice of $\eta$ to satisfy $0<1-\eta p(c-2)(\mu+\lambda)<1$, this term is always positive. Hence, the right hand side for Equation (12) is always positive.
> - **Equation 13**: With the Subsampled Gaussian Mechanism added during the aggregation step, there inevitably exists the term $\frac{\log(1/\delta)}{\epsilon^2}$. In other words, private SGD itself converges to a neighborhood around the optimum [2]. The second constant term comes from the mean-regularization term. In the non-private scenario, where the first constant term vanishes, the local objective can only converge to a neighborhood of the optimum. If this were *not* the case, the regularization term would asymptotically go to 0 as the number of communication rounds increases, in which case every personalized model would become the average model---losing the purpose of using MTL to train personalized models.
> - **Epsilon**: The purpose of Figure 2 is to demonstrate how the privacy parameter $\epsilon$ increases while the error decreases over communication rounds. We want to show that with weaker privacy guarantees (large value of $\epsilon$), we are able to get stronger utility performance (small value of loss). In practice, these larger values of $\epsilon$ would offer little privacy protection, but the figure itself helps to characterize the privacy-utility trade-off.
> - **Comparison to DP**: Although we agree with your intuition that MTL should perform better relative to global baselines in many scenarios (indeed, this has been demonstrated by numerous works in FL), we are not aware of any work that has considered how to ensure task/client-level privacy for these objectives, which is a major contribution of our work. Prior to our work, it was not clear (1) how to formally ensure task-level privacy in MTL, and (2) whether MTL methods would still perform better than global baselines when privacy was considered. We wish to re-emphasize that it is not possible to use traditional global DP with the MTL formulation, which is why we focus on JDP. We compare our private MTL formulation with traditional, global baselines in Section 5.2.

---

> > ### Author Response · Authors · 2021-11-17
> > **Response fBjV (continued)**
> >
> > - **Theory/Experiments**: As mentioned above, ours is the first work we are aware of to formalize task/client-level privacy for commonly used multi-task learning objectives. Theoretically, we provide an algorithm for private MTL and show that it provably achieves $(\epsilon,\delta)$-JDP in this setting. We also derive an extensive set of convergence analyses for both convex and non-convex objectives that take into account practical considerations of federated learning related to communication-efficiency and partial participation. As discussed in our paper, even in non-private settings, we are not aware of any work that has analyzed the convergence guarantees of commonly-used MTL objectives with these practical communication constraints. Empirically, we explore the efficacy of our approach on commonly-used large-scale FL benchmarks. We specifically show for the first time that it is possible for MTL methods to retain accuracy benefits over global baselines in FL while maintaining client/task-level privacy. If there are any additional experiments you feel are still warranted, we would greatly appreciate these suggestions.
> >
> > [1] Hsu, J., Huang, Z., Roth, A., Roughgarden, T., and Wu, Z. S.. Private matchings and allocations. SIAM Journal on Computing, 45(6), 1953-1984, 2016.
> >
> > [2] Bassily, R., Smith, A., and Thakurta, A. Private empirical risk minimization: Efficient algorithms and tight error bounds. Annual Symposium on Foundations of Computer Science, 2014.

---

> > > ### Comment · Reviewer_fBjV · 2021-11-20
> > > **Why is DP not applicable?**
> > >
> > > I don't fully understand the argument of DP not being applicable in this setting. In a non-federated setting, when you are training a model, you still want your model to not depend on any particular example too strongly. But that doesn't mean the model makes random / uninformed predictions. Maybe I am missing something here, can you shed more light on this?

---

> > > > ### Author Response · Authors · 2021-11-20
> > > > **Response**
> > > >
> > > > Yes, we are happy to explain this in more detail. The main reason that we can't use traditional DP directly is that we wish to enforce **task-level (not example-level) privacy, in a setting where each task produces its own separate model**. For example, this scenario has become commonplace in federated learning, where state-of-the-art MTL objectives produce a separate, personalized model for each client, yet we desire to preserve client-level privacy.
> > > >
> > > > As we discuss in Sections 1 and 3, the reason that this setting is incompatible with traditional DP is that (informally), DP states that the outputs (in this case, $m$ models, one for each client), shouldn't depend very much on any of the inputs (in this case, the 'inputs' are each client's **entire training dataset**). Unfortunately, client $k$'s model clearly can (and should) depend directly on at least some of client $k$'s data. If we forced this to **not** be true as in traditional DP, by definition, the predictions learned by model $k$ will be uninformed by/will not rely on any of the underlying data for task $k$. Said another way, for the application of FL, there is no reason to try to learn a personalized model for client $k$ when the model can't rely on any of the local data on client $k$. We provide a visualization of this issue in Figure 1.
> > > >
> > > > Although JDP was initially proposed as a privacy definition for solving problems in game theory and mechanism design, we are the first to establish the use of JDP to provide meaningful privacy guarantees when solving MTL.  In particular, JDP makes a natural relaxation, which is that for each task $k$, the set of output predictive models **for all other tasks except $k$** is insensitive to $k$'s private data. This formulation (along with the method/analyses we develop) allows us to still provide strong client-level privacy guarantees for applications like federated learning while also allowing state-of-the-art MTL objectives to be used. We believe that our initial work formalizing task-level privacy for mean-regularized MTL will be an important stepping stone for additional works exploring privacy in the context of MTL.

---

> > > > > ### Comment · Reviewer_fBjV · 2021-11-20
> > > > > **Some questions regarding convergence and baselines**
> > > > >
> > > > > Okay, thanks for clearing up th confusion regarding DP. I see now what you mean when you say DP isn't applicable in the way you have defined the problem.
> > > > >
> > > > > Regarding convergence, following up on the comment about last two terms - I agree that the absolute value of the objective can't go to zero, because of the way it has been defined (with the regularizer), but isn't this paper considering the difference between the value at iterate k, t and the best possible value? I believe there is some added confusion because the way empirical loss is defined in the paper seems to be a sum instead of an average, so all the terms might be missing a divide by n_k in the traditional sense. Even assuming this, the last term seems to be increasing with m, \eta being generally quite small is also concerning, and the value of B can be quite large if i understand correctly?
> > > > >
> > > > > The experiments also don't compare against simple private federated learning baselines like doing private FedAvg training or private MAML training, and personalizing using local training using the private global model as the initial point. The baselines to which the current algorithm is compared seem too easy/uninformative.

---

> > > > > > ### Author Response · Authors · 2021-11-21
> > > > > > **Response**
> > > > > >
> > > > > > **[Convergence analysis]**: Our convergence analysis studies the difference between $f_k(w_k^t;\bar{w}^t)$ and $f_k(w_k^*;\bar{w}^*)$ in the convex case and the gradient norm $\|\nabla_{w_k^t}f_k(w_k^t;\bar{w}^t)\|$ in the non-convex case. Note that in Theorem 3, the constant term does not increase with $m$; it is $\mathcal{O}(B)$. In Theorem 5, you are correct that we should not see that the last term increases with $m$, and we are sorry for the confusion. In particular, $\Delta_t$ in the Theorem 5 statement should be defined as $\sum_{k=1}^mf_k(w_k^t;\bar{w}^t)-f_k(w_k^*;\bar{w}^*)$. This term is first defined in Equation 105 and the summation should be included for $\Delta_t$ in the statement of Theorem 5. We have corrected this in the most recent revised PDF. The bound can therefore be simplified by dividing by $m$ on both sides, such that in the non-private case, $\frac{1}{m}\sum_{k=1}^mf_k(w_k^t;\bar{w}^t)-f_k(w_k^*;\bar{w}^*)$ is upper bounded by a term that diminishes with $T$ and a constant term that is $O(B)$, which also does not increase with $m$. As a result, the constant term in both convex and non-convex cases does not increase with $m$. Thank you for bringing this to our attention.
> > > > > >
> > > > > > Similar to our response to Reviewer T1p3, we provide a simple example to show that the ‘error’ present in the last term is expected. Assume that we have $m$ different clients/tasks, each with local data $x_i$ and local model $w_i$. The mean-regularized MTL objective for this problem would be $\frac{1}{m}\sum_{i}(x_i-w_i)^2+\frac{1}{m}\sum_i(w_i-\bar{w})^2$, which is greater than $\frac{1}{2m}\sum_i(x_i-w_i+w_i-\bar{w})^2 = \frac{1}{2m}\sum_i(x_i-\bar{w})^2$.  Note that this lower bound neither converges to 0 as $\bar{w}$ changes over time nor diminishes with increasing $m$. This result is therefore expected/standard, and is in line with other previous works that study the same objective but with different solvers, where you also see similar dependencies on \eta and B [1].
> > > > > >
> > > > > > **[Empirical baselines]**: Regarding our empirical study, the baselines that you describe are in fact exactly the baselines that we compare against. As you mention, the current de facto in federated settings is to train a global model with client-level DP using FedAvg. We compare against this baseline (and label it ‘global’) in Section 5.2 (please see Figure 3). These results show that when MTL performs better than the global baseline in non-private settings, we retain these benefits when privacy is added. Interestingly, we also see that even in scenarios where MTL is similar to the global baseline in non-private settings (e.g., the FEMNIST dataset), utility benefits exist when using private MTL relative to private FedAvg (i.e., ‘global’).
> > > > > >
> > > > > > We also compare to the baseline of personalization via local fine-tuning in Section 5.3. Since fine-tuning can be applied either to the global model or the MTL models, we consider the effect of fine-tuning both approaches. Even when considering fine-tuning, using private MTL models as a starting point rather than a global model performs better in most scenarios (and particularly in settings with greater privacy, i.e., small epsilon).
> > > > > >
> > > > > > [1]. F. Hanzely and P. Richtárik. Federated learning of a mixture of global and local models. arXiv preprint arXiv:2002.05516, 2020.

---

> > > > > > > ### Author Response · Authors · 2021-12-05
> > > > > > > **Response (follow up)**
> > > > > > >
> > > > > > > Dear Reviewer:
> > > > > > >
> > > > > > > Thanks again for your detailed comments and suggestions. We would like to check to see whether our response has adequately resolved your concerns? If not, we would appreciate it if you could let us know whether there are any further concerns that we can discuss. In particular, we hope that we have clarified our convergence results, and we wish to underscore that the private FL baselines you bring up are the ones that we have already explored in our work.

---

### Author Response · Authors · 2021-11-17
**Response to all reviewers:**

We thank all reviewers for their time and feedback on our submission. We have submitted a revised PDF of our submission with edits highlighted in red. We first address two shared points below and then respond to specific reviewer comments.

**[Contributions]**: Our work targets a critical missing link in practical applications of federated learning. Numerous works have demonstrated that multi-task learning models provide not only more accurate predictions in federated learning [1,2,3,4,5,6,7], but also have other benefits, including fairness and robustness properties [8]. From a practical point of view, it is therefore important to enable MTL for federated learning applications. However, there is a major gap here, as privacy is a first-order concern in FL, and it is unclear how to ensure client-level/task-level privacy while learning MTL models. As we discuss, using DP for MTL is not an option. Our work therefore bridges this gap by developing JDP methods for MTL---enabling state-of-the-art MTL objectives to be deployed in privacy-sensitive applications that were otherwise infeasible. While we focus on federated learning, we note that our approaches are applicable more generally to any practical scenario where it is desirable to use multi-task learning while ensuring the privacy of each underlying task.

**[DP vs JDP in MTL]**: Two reviewers alluded to seeing a baseline comparing MTL with DP to MTL with JDP. We wish to clarify that the notion of traditional, task-level DP is incompatible with MTL, which is why we did not run this ablation study or consider this as a baseline. In particular, as we discussed in Section 1, Figure 1, and Section 3.2, traditional DP requires that the output of a randomized algorithm be insensitive to changes in the input data. In the MTL setting, this would require that the set of all models be insensitive to changes on any client’s private data. In other words, to protect task-level privacy, traditional DP would force the model for a client’s task to be insensitive to changes in its own dataset---essentially resulting in completely random / uninformed predictions. As traditional DP is not an option, we instead propose JDP as an appropriate formalization for protecting task-level privacy of an MTL algorithm.

[1]. V. Smith, C. Chiang, M. Sanjabi, and A. Talwalkar. Federated multi-task learning. NeurIPS, 2017.

[2]. Y. Deng, M. M. Kamani, and M. Mahdavi. Adaptive personalized federated learning. arXiv preprint arXiv:2003.13461, 2020.

[3]. A. Ghosh, J. Chung, D. Yin, and K. Ramchandran. An efficient framework for clustered federated learning. NeurIPS, 2020.

[4] Hanzely, F., Hanzely, S., Horvath, S., and Richtarik, P. Lower bounds and optimal algorithms for personalized federated learning. NeurIPS, 2020.

[5] Dinh, C. T., Tran, N. H., and Nguyen, T. D. Personalized federated learning with moreau envelopes. NeurIPS, 2020

[6]. F. Hanzely and P. Richtárik. Federated learning of a mixture of global and local models. arXiv preprint arXiv:2002.05516, 2020.

[7]. Y. Mansour, M. Mohri, J. Ro, and A. T. Suresh. Three approaches for personalization with applications to federated learning. arXiv preprint arXiv:2002.10619, 2020.

[8]. T. Li, S. Hu, A. Beirami, and V. Smith. Ditto: Fair and robust federated learning through personalization. ICML 2021.

---

### Decision · Program_Chairs · 2022-01-20

**Decision:**

Reject

**Comment:**

Many problems in machine learning rely on multi-task learning (MTL), in which the goal is to solve multiple related machine learning tasks simultaneously. In this work, authors formalize notions of task-level privacy for MTL via joint differential privacy (JDP). They propose an algorithm for mean-regularized MTL, an objective commonly used for applications in personalized federated learning, subject to JDP. Then analyze objective and solver, providing certifiable guarantees on both privacy and utility. The main results, namely the convergence rate results, are hard to parse and hard to interpret. For example, as one reviewer pointed out, it is bounded below by a constant which is not properly explained. Further, comparisons to the literature in user-level privacy (which is equivalent as the task-level privacy) is not provided enough. Significant improvement in the presentation of the main results, along with an interpretable explanation of the contribution, is necessary for this manuscript.